



**Study of electron spectral diffusion process under DNP conditions by ELDOR spectroscopy focusing on the $^{14}$N Solid Effect**

Marie Ramirez Cohen, Akiva Feintuch, Daniella Goldfarb*, Shimon Vega*

[1]Department of Chemical Physics, Weizmann Institute of Science, Rehovot, Israel.

*Corresponding authors: Daniella Goldfarb (daniella.goldfarb@weizmann.ac.il),
Shimon Vega (shimon.vega@weizmann.ac.il)

**Abstract** Electron spectral diffusion (eSD) plays an important role in solid state, static DNP with polarizers having in-homogeneously broadened EPR spectra, such as nitroxide radicals. It affects the electron spin polarization gradient within the EPR spectrum during microwave irradiation and thereby determines the effectiveness of the DNP process via the so called indirect cross effect (iCE) mechanism. The electron depolarization profile can be measured by Electron-Electron Double Resonance (ELDOR) experiments and a theoretical framework for deriving eSD parameters from ELDOR spectra and employing them to calculate DNP profiles has been developed. The inclusion of electron depolarization arising from the $^{14}$N Solid Effect (SE) has not yet been taken into account in this theoretical framework and is the subject of the present work. The $^{14}$N SE depolarization was studied using W-band ELDOR of a 0.5 mM TEMPOL solution, where eSD is negligible, taking into account the hyperfine interaction of both $^{14}$N and $^{1}$H nuclei, the long microwave irradiation applied under DNP conditions and electron and nuclear relaxation. The results of this analysis were then used in simulations of ELDOR spectra of 10 and 20 mM TEMPOL solutions, where eSD is significant using the eSD model and the SE contributions were added ad-hoc employing the $^{1}$H and $^{14}$N frequencies and their combinations, as found from the analysis of the 0.5 mM sample. This approach worked well for the 20 mM solution where a good fit for all ELDOR spectra recorded along the EPR spectrum was obtained and the inclusion of the $^{14}$N SE mechanism improved the agreement with the experimental spectra. For the 10 mM solution, simulations of the ELDOR spectra recorded along the $g_z$ position gave a lower quality fit than for spectra recorded in the center of the EPR spectrum, suggesting that the simple approach used to the SE of the $^{14}$N contribution, when its contribution is high, is lacking as the anisotropy of its magnetic interactions has not been considered explicitly.



# 1 Introduction

It has been recently recognized that electron spectral diffusion (eSD) plays a significant role in dynamic nuclear polarization (DNP) under static conditions(Hovav et al., 2015a; Leavesley et al., 2017) . It  affects the electron spin polarization gradient within the EPR spectrum as a consequence of microwave irradiation and thereby determines the effectiveness of the DNP process via the so called indirect cross effect (iCE) mechanism(Hovav et al., 2015a).  This is particularly relevant in the case of nitroxide radicals, the EPR spectra of which are in-homogeneously broadened in frozen solutions, at concentrations of 20-40 mM used in DNP applications. Hovav *et al* (Hovav et al., 2015b, 2015a), Siaw *et al*(Siaw et al., 2014) and Shimon *et al*(Shimon et al., 2012, 2014) observed that during constant microwave (MW) irradiation there exists an optimal radical concentration that leads to a maximum in the DNP enhancement. At this concentration the inter-electron spin dipolar interaction is sufficiently strong to generate a polarization gradient that favors an efficient iCE enhancement mechanism, while at higher concentrations the spectral diffusion  saturates large parts of the EPR spectrum and spin temperature effects can be expected(Kundu et al., 2018a, 2018b). To monitor directly the electron depolarization during MW irradiation,  Hovav *et al*(Hovav et al., 2015b) measured the ELDOR signals of frozen TEMPOL solutions under static DNP conditions,  as a function of TEMPOL concentration, sample temperature and MW irradiation time. Furthermore, they developed a model (called the eSD model) that describes the depolarization process. This model is based on rate equations for the electron polarizations along the EPR spectrum, taking into account an exchange process between polarizations, in addition to the saturation effects of the MW irradiation and the spin-lattice relaxation. This eSD model introduces a fitting parameter $\Lambda^{eSD}$ that defines the strength of the polarization exchange rate leading to the spectral diffusion within the EPR spectrum. Using this eSD model, experimental ELDOR spectra could be satisfactorily simulated and thus provide a feasible analytical description of the eSD process. Subsequently, it was demonstrated that once the polarization gradient within the EPR spectrum has been determined via the eSD model simulations, the lineshape of the associated DNP spectrum could be reproduced  taking into account the polarization differences between all electron pairs satisfying the CE condition(Hovav et al., 2015a). This approach was also implemented by Leavesley *et al,* (Leavesley et al., 2017) when they explored the eSD process and its influence on the DNP efficiency

at a magnetic field of 7T. They also considered the effects of variations in the radical
concentration, temperature and MW power on the [1]H-DNP spectra. Furthermore,
Kundu *et al.* used the eSD model to quantify the dependence of the electron polarization
exchange parameter $\Lambda^{eSD}$ on radical concentration and temperature[7].
To justify the rather phenomenological eSD model, Kundu *et al*(Kundu et al., 2018a,
2018b) performed quantum mechanical based calculations of the spin evolution and
associated EPR spectra of the electron spins in  dipolar coupled small spin systems
under DNP conditions. In the case of weak dipolar coupling constants and adding cross
relaxation(Hwang and Hill, 1967; Kessenikh et al., 1964) to the ELDOR calculations
the results were similar to those obtained  using the eSD model. In the case of strong
dipolar couplings a Thermal Mixing mechanism in the rotating frame could provide the
calculated EPR spectra under MW irradiation.(Abragam, 1961; de Boer, 1976;
Borghini, 1968; Goldman, 1970; Provotorov, 1962; Wenckebach, 2016; Wollan, 1976)
These studies also contributed to the validity of the iCE model in the weak and the
strong coupling regime.
In addition to the CE mechanism, leading to the main nuclear signal enhancements at
high radical concentrations, the Solid Effect (SE) process also influences these
enhancements. This process contributes to the signal enhancements, but in addition
causes some electron depolarization that in turn can influence the CE enhancement
process(Hovav et al., 2015b; Leavesley et al., 2018). When nitroxide radicals are used
as DNP polarizers, these SE depolarization effects arise from [1]H and [14]N nuclei
hyperfine interactions(Kundu et al., 2018b; Leavesley et al., 2017). The SE induced
polarization depletions are highly evident  in ELDOR spectra at concentrations that are
below the usual concentration used for DNP, but their influence is observed also at
concentrations around 20 mM, which are relevant for DNP(Harris et al., 2011; Lilly
Thankamony et al., 2017). As the ELDOR lineshapes are simulated for the
determination of the $\Lambda^{eSD}$ constants, the SE effects should be taken into account in the
eSD model to ensure the extraction of their correct value. The purpose of this study is
to account explicitly for the effects of the SE mechanism on ELDOR lineshapes for
nitroxides and to explore its influence on the extraction of the $\Lambda^{eSD}$ parameter at
concentrations relevant for static DNP.





We started this study by measuring ELDOR spectra of a 0.5 mM TEMPOL in DMSO
frozen solution, in which the SE is the sole mechanism of depolarization, as the spectral
diffusion mechanism is negligible. To analyze these ELDOR spectra we established a
theoretical framework that accounts for all $^{14}$N-SE and $^{1}$H-SE depletions observed in
these spectra. For this low concentration, the ELDOR spectrum is identical to the
ELDOR detected NMR (EDNMR) spectrum of nitroxide, which has already been
studied and simulated in the past(Cox et al., 2017; Florent et al., 2011; Kaminker et al.,
2014; Nalepa et al., 2014, 2018). Yet, there is one major difference: Under EDNMR
conditions, where resolution is of prime interest, the MW irradiation period is short, in
the microsecond range, and therefore relaxation processes play a limited role during
that irradiation. However, under DNP conditions the duration of the irradiation is in the
range of milliseconds or longer and the electron and nuclear relaxation processes
influence the magnitude of the depolarization. A second, more technical, difference is
that in a full two dimensional (2D) EDNMR spectrum the EPR dimension is usually
obtained by stepping the magnetic field(Kaminker et al., 2014; Ramirez Cohen et al.,
2017), unless chirped pulses are being used(Wili and Jeschke, 2018), while 2D ELDOR
maps in the context of DNP are obtained by stepping the frequency. Finally, so far
contributions from different nuclei in the EDNMR spectra were taken into account by
superimposing their individual spectra(Wang et al., 2018), ignoring the contributions
of combination frequencies(Tan et al., 2019). Here we also account for $^{14}$N-$^{1}$H
combination lines in the ELDOR spectrum.
After analyzing the 0.5 mM spectrum, we proceeded to 10 and 20 mM TEMPOL
solutions, where spectral diffusion becomes significant. We measured their ELDOR
spectra and analyzed them employing  the eSD model(Hovav et al., 2015b), taking into
account the SE mechanism through an ad-hoc inclusion of the $^{14}$N and $^{1}$H frequencies.

## 2. Methods and Materials


### 2.1 Sample preparation


Samples of 2-3 μl in 0.6mm ID x 0.84 mm quartz tubes, with 0.5, 10 and 20 mM
TEMPOL dissolved in a solution of DMSO/H$_2$O (1:1 v/v), were degassed by a *Freeze-*
*Pump-Thaw* procedure and fast frozen with liquid Nitrogen. TEMPOL and DMSO were
both purchased from Sigma Aldrich and used as is.



## 2.2 Spectroscopic measurements

All measurements were carried out on our W-band (95 GHz, 3.4 T) homebuilt EPR spectrometer(Goldfarb et al., 2008; Mentink-Vigier et al., 2013) at 20 K.

Echo-detected EPR (ED-EPR) spectra were measured using the pulse sequence $\pi/2$-$\tau$-$\pi$ -$\tau$-echo with $\tau$=600 ns, while increasing the magnetic field stepwise from 3370 to 3395 mT, with a 2 ms repetition time. The pulse lengths were 100 ns for the $\pi/2$ pulse and 200 ns for the $\pi$ pulse, optimized at the detection frequency (94.90 GHz).

Electron spin-lattice relaxation times $T_{1e}$ were measured by saturation recovery experiments, using a long MW saturation pulse of 30 ms at different positions within the EPR spectrum and the echo was detected with low MW power as typical for DNP using two 300 ns pulses. The $T_{1e}$ curves were analyzed using a superposition of two exponential functions with time constants $t_1$ and $t_2$, with the slow (and major) component assigned to $T_{1e}$.

ELDOR spectra were measured at different detection frequencies along the EPR line. The pulse sequence is shown in Figure 1. The spectrometer was set to low power as typical for DNP and therefore detection was performed by the sequence $\alpha$-$\tau$- $\alpha$ -$\tau$-echo where $\alpha$ is a flip angle of less than $\pi/2$, While for EPR applications ELDOR is carried out at a fixed detection frequency and the magnetic field is varied to access different regions in the EPR spectrum, here we kept the field constant and varied the detection frequency to access the spectrum width as done for DNP applications. To carry out these ELDOR measurements, we increased the bandwidth of the cavity to accommodate the full spectrum of TEMPOL (approx. 500 MHz). The cavity resonance was tuned to 94.80 GHz. For the 0.5 mM sample ELDOR spectra (40 in total) were recorded as a function of the pump frequency, which was varied from 94.3 GHz to 95.3 GHz. To obtain two-dimensional ELDOR data the ELDOR spectrum was measured at

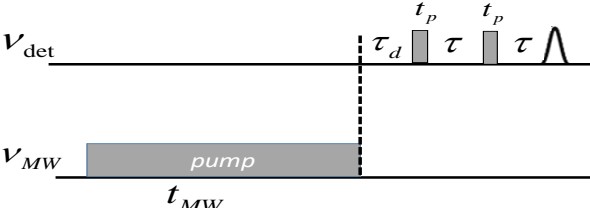

Figure 1. ELDOR pulse sequence, where $v_{det}$ is the detection frequency and $v_{MW}$ is the frequency of the pump pulse and $t_{MW}$ is the duration of the pump pulse.



different detection frequencies in intervals of 10 MHz from 94.55 GHz to 94.95 GHZ,
which covers most of the EPR spectrum. The amplitude of the pump pulse, $\nu_1$, was 0.5
MHz, as determined by a nutation experiment at 94.8 GHz, corresponding to an
inversion pulse of 1μs. The experimental parameters for the ELDOR experiments are
listed in **Table 1**.
**Table 1**. Parameters used in EDNMR experiment for 0.5, 10 and 20 mM radical concentration
(see Fig. 1)

| $t_p$ | $T$ | $t_{MW}$ | Repetition time | $\tau_d$ |
|-------|-----|----------|-----------------|----------|
| 300 ns | 600 ns | 10 ms | 20 ms | 6 μs |

## 3 Simulations

### 3.1 Low radical concentrations

*The Hamiltonian and the allowed transition*
In an effort to analyze the ELDOR spectra of the 0.5 nm TEMPOL solution we rely on
quantum mechanical based calculations considering the spin evolution of a three-spin
system consisting of an electron spin, S=1/2, coupled to a single [1]H nucleus and a
single [14]N nucleus. Simulations of these ELDOR spectra were performed using a
modified version of the computer code developed by Kaminker *et al*(Kaminker et al.,
2014) for a two-spin system; one electron spin and one [14]N nucleus. The simulated
ELDOR spectra comprise of EPR signals calculated at fixed detection frequency
positions $\nu_{det} = \omega_{det} / 2\pi$ as a function of the of pump pulse frequency, $\nu_{MW} = \omega_{MW} / 2\pi$
. During these calculations, we had to take into account the fact that the duration of the
MW irradiation in DNP experiments $t_{MW}$ is much longer than commonly used in
EDNMR spectroscopy (ms vs μs range, respectively). For such long irradiation times
the three-spin calculations cannot account for the experimental spectral observations,
mainly due to the fact that the real spin system is more extended than only three spins
because of the many coupled protons present in the sample. Accordingly, without
extending the number of spins in our model we had to modify Kaminker's procedure
to reproduce the experimental observations, as will be discussed here below.
The three-spin system is described by the following spin Hamiltonian in the MW
rotating frame, assuming the high field approximation:



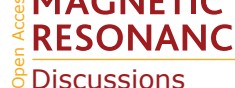

$$\hat{H}_{\theta,\varphi} = \Delta\omega_e \hat{S}_z - \omega_N \hat{I}_{zN} - \omega_H \hat{I}_{zH} + A_{zz}^H \hat{S}_z \hat{I}_{zH} + A_{zz}^N \hat{S}_z \hat{I}_{zN}$$
$$+ (A_H^+ \hat{I}_H^+ + A_H^- \hat{I}_H^-)\hat{S}_z + (A_N^+ \hat{I}_N^+ + A_N^- \hat{I}_N^-)\hat{S}_z + \hat{I}_N \cdot \tilde{Q} \cdot \hat{I}_N \qquad (1)$$

where
$$\Delta\omega_e \hat{S}_z = (\mu_B B_0 g_{eff}(\theta,\varphi) - \omega_{MW})\hat{S}_z . \qquad (2)$$
In Eq. 1 we neglected the dipolar interaction between the nuclei. $\Delta\omega_e$ is the off-
resonance electron frequency, $B_0$ is the strength of the external magnetic field, pointing
along the z-axis of the laboratory frame, and $g_{eff}(\theta,\varphi)$ is the effective $g$-tensor
parameter for a specific orientation of the magnetic field with respect to the principle
axis system of the $g$-tensor, given by the polar angles $\theta$ and $\varphi$. The $g$ tensor used for
the calculation is $g$ = [2.0065, 2.0037, 1.9997], obtained by simulating, using
Easyspin(Stoll and Schweiger, 2006), the frequency domain EPR spectrum extracted
from the echo intensity of the ELDOR spectra with the pump pulse set far outside the
EPR spectrum (see Fig. S1 in ESI). The $g$-values obtained differ from those reported
by Florent *et al*(Florent et al., 2011) ($g$ = [2.00988, 2.00614, 2.00194]) and this seems
to be due to a systematic error of 4 mT in the determination of the external magnetic
field. The Larmor frequencies of $^1$H and $^{14}$N are $\omega_H = 2\pi\nu_H$ and $\omega_N = 2\pi\nu_N$,
respectively. In the EPR high field approximation the terms that contribute to the
hyperfine interaction are the secular and pseudo-secular terms with coefficients
$(A_{zz}^H, A_H^\pm)$ for $^1$H and $(A_{zz}^N, A_N^\pm)$ for $^{14}$N, where $A_\pm^K = A_{zx}^K \pm iA_{zy}^K$, $K = $ H,N. In the case of
$^{14}$N the hyperfine tensor contains an isotropic contribution $a_{iso}^N \neq 0$ in addition to the
anisotropic tensor elements $[a_{ZZ}^K, a_{XX}^K, a_{YY}^K]$, where $X$, $Y$ and $Z$ are its principle axes.
Assuming that the two anisotropic hyperfine interactions are of axial symmetry (i.e.
$a_{XX}^K = a_{YY}^K = -1/2 a_{ZZ}^K$) and that their major principal axes coincide with that of the $g$-
tensor,   the   hyperfine   coefficients   of   $\hat{H}_{\theta,\varphi}$   become
$A_{zz}^K \equiv A_{zz}^K(\theta) = a_{iso}^K + \frac{1}{2}a_{ZZ}^K(3\cos^2\theta-1)$ and $A_\pm^K \equiv A_\pm^K(\theta) = \frac{3}{2}a_{ZZ}^K\cos\theta\sin\theta$ (Schweiger
and Jeschke, 2001). In the case of TEMPO, the isotropic $^{14}$N contribution is
$a_{iso}^N = 44$ MHz  and the anisotropic value is $-a_{ZZ}^N = 55$ MHz. The $^1$H hyperfine value
was taken as is $a_{ZZ}^H = 3$ MHz. Finally, the $^{14}$N nuclear quadrupole interaction is also
included in the spin Hamiltonian. Here we used the principal values of the quadrupole



tensor    obtained    by    Florent    *et    al*(Florent    et    al.,    2011),
$(Q_{XX}, Q_{YY}, Q_{ZZ}) = (0.48, 1.29, -1.77)$ MHz, and again assumed that its principal axes
coincides with those of the $g$- tensor.
The MW irradiation Hamiltonian in the rotating frame is defined as
$$\hat{H}_{MW} = \omega_1 \hat{S}_x \ .$$ (3)
At the start of all our simulations, the Hamiltonian for each set of $(\theta, \varphi)$ angles is
represented in matrix form, in the twelve product states of the basis sets in the laboratory
frame $|\chi_e\rangle$, $|\chi_H\rangle$ with $\chi_{e,H} = \alpha, \beta$ and $|\chi_N\rangle$ with $\chi_N = +1, 0, -1$, and diagonalized
according to
$$\hat{\Lambda}_{\theta,\varphi} = \hat{D}_{\theta,\varphi}^{-1} \hat{H}_{\theta,\varphi} \hat{D}_{\theta,\varphi} \ .$$ (4)
$\hat{D}_{\theta,\varphi}$ is the diagonalization matrix and $\hat{\Lambda}_{\theta,\varphi}$ is the diagonal matrix consisting of the
eigenvalues $E_i^{\theta,\varphi}$, in frequency units, corresponding to the 12 eigenstates $|\lambda_i^{\theta,\varphi}\rangle$ with
$i = 1,,,12$. The EPR transition probabilities between levels $|\lambda_i^{\theta,\varphi}\rangle$ and $|\lambda_j^{\theta,\varphi}\rangle$ are :
$$P_{i,j}^{\theta,\varphi} = 2\left|\langle \lambda_i^{\theta,\varphi}|D_{\theta,\varphi}^{-1} \hat{S}_x D_{\theta,\varphi}|\lambda_j^{\theta,\varphi}\rangle\right|^2 .$$ (5)
When $|Q_{ZZ}| < \omega_N < \frac{1}{2}a_{ZZ}^N, a_{iso}$, the $\omega_n I_z^N$ term in all $H_{\theta,\varphi}$ Hamiltonians has little influence
on the form of the eigenstates, which are products of the electron states $|\chi_e\rangle$ with the
eigenvalues $m_e = \pm 1/2$, the hyperfine mixed proton states approximately equivalent to $|\chi_H\rangle$
with $m_H \approx \pm 1/2$ and the nitrogen states $|\chi_N\rangle$, mainly determined by the hyperfine interaction
terms in $H_{\theta,\varphi}$ with $m_N \approx +1, 0, -1$. As a result we can easily recognize six "allowed"
transition with frequencies $v_{(i,j)_a}(\theta, \varphi) = (E_i^{\theta,\varphi} - E_j^{\theta,\varphi})$ that correspond to EPR transitions
$(i-j)_a$, with $\Delta m_e = \pm 1$, $\Delta m_H \approx 0$ and $\Delta m_N \approx 0$ and thus $P_{i,j}^{\theta,\varphi} \approx 1$. We note that for
orientations along the X,Y axis, the $^{14}$N hyperfine interaction is close to $\omega_N$ and therefore
$P_{i,j}^{\theta,\varphi} < 1$. Figure 2 presents a schematic energy level diagram of the three-spin system
for an arbitrary set of angles $(\theta, \varphi)$. The six allowed transitions are indicated by red
arrows. For one of these transitions the corresponding homonuclear "single quantum"



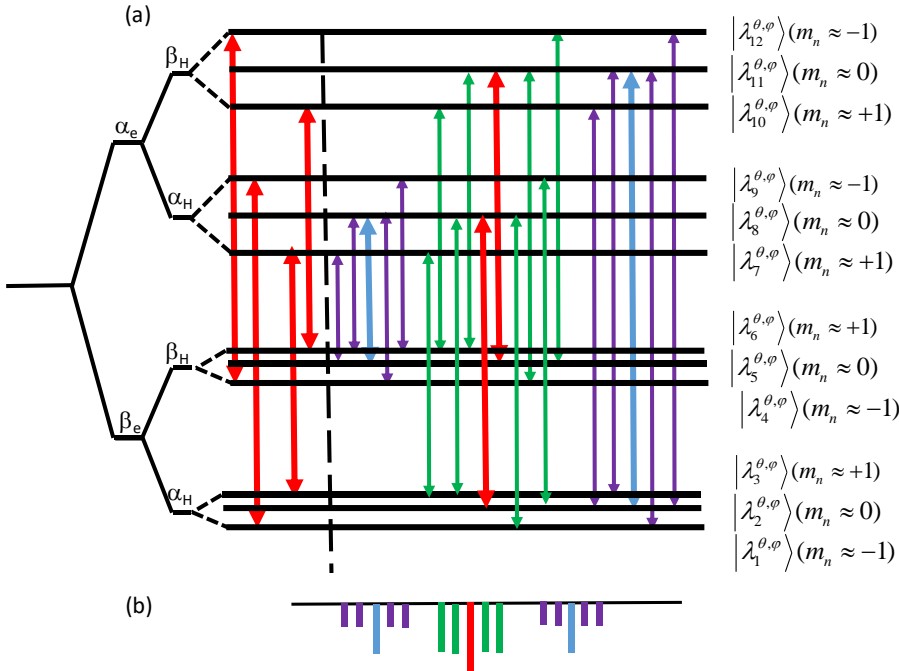

**Figure 2.** (a)A schematic energy level diagram of the three spin system with angles θ,φ , corresponding to an allowed transition. The eigenstates $\left|\lambda_i^{(\theta,\varphi)_{det}}\right\rangle$ are characterized by their $m_N$ (** small n) values and product states of $\left|\chi_e\right\rangle$, $\left|\chi_H\right\rangle$ and $\left|\chi_N\right\rangle$ . The energy level differences $\nu_e$ and $\nu_H \pm A_H$ are scaled arbitrarily. On the left of the energy level diagram the allowed transitions (3-7), (6-10) ,(1-9), (4-12) are indicated by the red arrows. On the right the red arrows correspond to the allowed transition between the states with sub-indexes (2-8) and (5-11). The nitrogen forbidden transitions (2-9), (2-7), (4-11) and (6-11) are assigned by the green arrow and the proton forbidden transitions (2-11) and (5-8) by the blue arrows. The purple arrows indicate the combined proton-nitrogen transitions. (b) A schematic presentation of the ELDOR spectrum of the allowed overlapping (2-8) and (5-11) transitions following the color coding of the arrows.

(SQ) forbidden transitions, with $\Delta m_H \approx \pm 1$ or $\Delta m_N \approx \pm 1$, are also indicated, in blue or
green, respectively. The heteronuclear "double-" and "zero quantum" (DQ and ZQ)
forbidden transitions, with $\Delta m_H \approx \pm 1$ and $\Delta m_N \approx \pm 1$, are shown in purple.
Using the Orisel function in Easyspin(Stoll and Schweiger, 2006), the values of $E_i^{\theta,\varphi}$
and $P_{i,j}^{\theta,\varphi}$ were calculated for a collection of 9609 sets of values of $(\theta,\varphi)$ and from them
all transition frequencies $\nu_{i,j}(\theta,\varphi)$ were determined. To choose which orientations of
the spin system contribute to the allowed EPR signal at a given $\nu_{det}$, we search for those
sets of angles $(\theta,\varphi)$ for which at least one allowed transition falls in the frequency

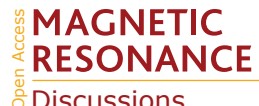

range $\nu_{det} - 3\text{MHz} \le \nu_{(i,j)_a}(\theta, \varphi) \le \nu_{det} + 3\text{MHz}$. This frequency span provides a
frequency bandwidth of 6 MHz for the detection pulse, estimated as the excitation
bandwidth for a detection pulse of 300 ns length. In addition, it can account for some
$g$- and hyperfine strain. This procedure generated a subset of selected $(\theta, \varphi)_{det}$ pairs for
each $\nu_{det}$, the size of which depends on the position of $\nu_{det}$ within the EPR spectrum.
After choosing a value for $\nu_{det}$ we simulated the ELDOR spectra of all crystal
orientations of the subset $(\theta, \varphi)_{det}$. The sum of these spectra are compared with the
measured ELDOR spectrum at that frequency. To obtain the individual ELDOR spectra
we calculated the EPR signal at $\nu_{det}$ after a long MW pump pulse as a function of the
frequency of this pulse, $\nu_{MW}$.
*The population rate equation*
To follow the evolution of the spin system during the long MW irradiation period, prior
to the EPR detection, it is sufficient to consider only the eigenstate populations $p_i^{\theta,\varphi}(t)$
of all $\left| \lambda_i^{\theta,\varphi} \right\rangle$, as described earlier (Hovav et al., 2010, 2015b). The rate equation during
the MW irradiation for these populations can be presented as

$$\frac{d}{dt} p_i^{\theta,\varphi} = \sum_{j=1,12} \{-R_{ij}^{\theta,\varphi} + W_{ij}^{\theta,\varphi}\} p_j^{\theta,\varphi}, \tag{6}$$

where $R_{ij}^{\theta,\varphi}$ are the elements of the 12x12 spin lattice relaxation matrix $\hat{R}_{\theta,\varphi}$ and $W_{ij}^{\theta,\varphi}$ are the
elements of the 12x12 MW rate matrix $\hat{W}_{\theta,\varphi}$. The relaxation matrix $\hat{R}_{\theta,\varphi}$ is equal to the sum of
the relaxation matrices $\hat{r}_{(ij)}^{\theta,\varphi}$ of all transitions $\{i-j\}$ with $E_j > E_i$. The non-zero matrix
elements of $\hat{r}_{(ij)}^{\theta,\varphi}$ are derived, assuming a linear field fluctuation causing $T_{1e}$:

$$r_{(ij),ii}^{\theta,\varphi} = -\frac{1}{T_{1,ij}} \frac{1}{(1+\eta_{ij})} \quad ; \quad r_{(ij),ij}^{\theta,\varphi} = \frac{1}{T_{1,ij}} \frac{\eta_{ij}}{(1+\eta_{ij})}$$
$$r_{(ij),ji}^{\theta,\varphi} = \frac{1}{T_{1,ij}} \frac{1}{(1+\eta_{ij})} \quad ; \quad r_{(ij),jj}^{\theta,\varphi} = -\frac{1}{T_{1,ij}} \frac{\eta_{ij}}{(1+\eta_{ij})} \tag{7a}$$

and

$$\frac{1}{T_{1,ij}} = \frac{\left| \left\langle \lambda_i^{\theta,\varphi} \left| \hat{S}_x \right| \lambda_j^{\theta,\varphi} \right\rangle \right|^2}{T_{1e}} \tag{7b}$$



with $\eta_{ij}^{\theta,\varphi} = p_i^{\theta,\varphi;eq} / p_j^{\theta,\varphi;eq}$ being the ratio between the thermal equilibrium populations
and

$$\hat{R}_{\theta,\varphi} = \sum_{\{i-j\}} r_{(ij)}^{\theta,\varphi}. \tag{7c}$$

The elements of $\hat{W}_{\theta,\varphi}$ are equal to the sum of the $\hat{w}_{(ij)}^{\theta,\varphi}$ matrices with non-zero elements
that express the effective irradiation strength on each transition $(i-j)$ (Hovav et al.,

6   2010):

$$w_{(ij),ij}^{\theta,\varphi} = w_{(ij),ji}^{\theta,\varphi} = -w_{(ij),ii}^{\theta,\varphi} = -w_{(ij),jj}^{\theta,\varphi} = \frac{\omega_1^2 \left| \left\langle \lambda_i^{\theta,\varphi} \middle| \hat{S}_x \middle| \lambda_j^{\theta,\varphi} \right\rangle \right|^2 T_{2mw}}{1 + 4\pi^2 \left\{ \nu_{ij}^{\theta,\varphi} - \nu_{MW} \right\}^2 T_{2mw}^2} \tag{8a}$$

and

$$\hat{W}_{ij}^{\theta,\varphi} = \sum_{(i-j)} \hat{w}_{(ij)}^{\theta,\varphi}. \tag{8b}$$

Here $\omega_1$ is the MW amplitude (see Eq. 3). A transverse relaxation time $T_{2mw}$, which
determines the off-resonance efficiency of the irradiation, is introduced and for
simplicity is assumed to be the same for all transitions. Note that $T_{2mw}$ is not the
measured phase memory time, $T_M$, which can serve as a lower limit for $T_{2mw}$. After
choosing values for $T_{1e}$, $\omega_1$ and an irradiation time, it is possible to solve Eq. 6 and to
use the populations at the end of the irradiation to evaluate the EPR signals.
Setting the detection frequency at one of the allowed transition frequencies and
irradiating with a pump frequency that matches one of its associated forbidden
transitions (i.e, they share a common energy level) result in a depletion of the EPR
signal. The calculations show that the depletion can be very significant for pump pulses
on the order of tens of microseconds but disappears for irradiation periods of the order
of tens of milliseconds. Thus using Eq. 6 works well for calculating EDNMR spectra
for short pump pulses(Kaminker et al., 2014; Ramirez Cohen et al., 2017). However,
for extended periods of MW irradiation, longer than $T_{1e}$ as is applied in DNP, the
simulated ELDOR signals reveal very weak signal at the forbidden transition
frequencies. The reason for this is that for MW irradiations longer than $T_{1e}$, the SE spin
evolution of an electron-nuclear spin pair brings the electronic polarization back to its
equilibrium value. This is, however, in contrast to the experimental results where rather
intense lines were observed even for long irradiation. The reason for this discrepancy



is that in reality the electron spins are interacting with several equivalent coupled nuclei,
which transfer their polarization to the bulk via nuclear spin diffusion. This is
particularly true when many protons are present. Accordingly, to reproduce the
experimental results, while still employing our simplified three-spin system model,
requires modification of the simulation procedure as described next.
*iii. Modification of the rate equation*
In order to obtain from a three-spin calculations the observed EPR signal depletions
even after long irradiation periods, we modified the form of the MW rate matrix.
Realizing that an irradiation of one of the forbidden transitions, $(i-k)_f$ and $\underline{(k-j)_f}$,
causes a depletion of the population difference of an allowed transition, $(i-j)_a$, we
removed the four matrix elements of $\hat{w}^{\theta,\varphi}_{(ik)_f}$ and $\hat{w}^{\theta,\varphi}_{(kj)_f}$ from the $\hat{W}_{\theta,\varphi}$ matrix. This is
equivalent to removing the irradiation on the forbidden transitions, which in turn cause
the change in population difference of the allowed transition, $P^{\theta,\varphi}_{i,j}$. To re-introduce the
effect of the forbidden transitions on $P^{\theta,\varphi}_{i,j}$ of the allowed transitions, we added
reintroduced them as an artificial irradiation on the allowed one by adding them to the
four non-zero matrix elements of $\hat{w}^{\theta,\varphi}_{(ij)_a}$ : $\left\{ \hat{w}^{\theta,\varphi}_{(ik)_f} + \hat{w}^{\theta,\varphi}_{(kj)_f} \right\}_{(ij)_a}$. In this way we ensure
a depletion of the population difference of $(i-j)_a$, without the relaxation mechanism
cancelling it. Realizing that the depletion during the simulations is now dependent on
the value of $T_{i,ij}$, we introduce SE fitting parameters to adjust their values during
irradiation: on for the different forbidden proton, $a^{SE}_H$, nitrogen, $a^{SE}_N$, combined proton-
nitrogen, $a^{SE}_{HN}$ and even double quantum (DQ) nitrogen, $a^{SE}_{DQ-N}$ transitions. In this way
an irradiation on $(i-k)_f$ reproduced the experimentally observed signal depletions,
still taking into account the effective MW irradiation strengths, $\omega_1 \times \left\langle \lambda^{\theta,\varphi}_i \left| \hat{S}_x \right| \lambda^{\theta,\varphi}_k \right\rangle$,
and its original off resonance efficiency. Performing this procedure for all forbidden
transitions, the modified $\hat{W}_{\theta,\varphi}$ matrix contains only elements corresponding to the
allowed transitions $(i-j)_a$:

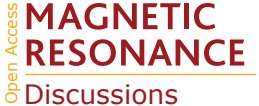

$$\hat{W}_{\theta,\varphi} = \sum_{\substack{6\ allowed \\ (i-j)_a}} \hat{W}^{\theta,\varphi}_{(ij)_a} \quad ;$$

$$\hat{W}^{\theta,\varphi}_{(ij)_a} = \hat{w}^{\theta,\varphi}_{ij} + a^{SE}_N \sum_{(ik)_N;(kj)_N} \left\{\hat{w}^{\theta,\varphi}_{(ik)_N} + \hat{w}^{\theta,\varphi}_{(kj)_N}\right\}_{(ij)_a} + a^{SE}_{DQ-N} \sum_{(ik)_{DQ-N};(kj)_{DQ-N}} \left\{\hat{w}^{\theta,\varphi}_{(ik)_{DQ-N}} + \hat{w}^{\theta,\varphi}_{(kj)_{DQ-N}}\right\}_{(ij)_a}$$

$$+ a^{SE}_H \sum_{(ik)_H;(kj)_H} \left\{\left\{\hat{w}^{\theta,\varphi}_{(ik)_H} + \hat{w}^{\theta,\varphi}_{(kj)_H}\right\}_{(ij)_a} + a^{SE}_{HN} \sum_{(kl)_{HN};(lk)_{HN}} \left\{\hat{w}^{\theta,\varphi}_{(kl)_{HN}} + \hat{w}^{\theta,\varphi}_{(lk)_{HN}}\right\}_{(ij)_a}\right\}$$

$$. \tag{9}$$

Here the sums over $k$ and $l$ of $(ik)_K$, $(kj)_K$, $(kl)_{KK'}$, $(lk)_{KK'}$ are restricted to the homo-nuclear and hetero-nuclear forbidden transitions only. After this modification it becomes possible to write for each allowed transition $(i-j)_a$ a 2x2 rate equation for the populations $p^{\theta,\varphi}_i(i)$ and $p^{\theta,\varphi}_j(t)$ with a rate matrix $(-\hat{r}_{(ij)_a} + \hat{W}_{(ij)_a})$.

The actual relaxation pathways in the spin system is influenced by all the elements of $\hat{R}_{\theta,\varphi}$ and as a result, an irradiation on one allowed transition can have a small effect on the populations of another allowed transition.(Kaminker et al., 2014) Our modification caused this effect to vanish in the simulations. To reintroduce it in our simulations we added to each $\hat{W}^{\theta,\varphi}_{(ij)_a}$ the MW rate matrices of the other transitions $\hat{W}^{\theta,\varphi}_{(kl)_a}$, while introducing an additional small fitting parameter $a_{a-a}$:

$$\hat{W}^{\theta,\varphi}_{(ij)_a} = \hat{W}^{\theta,\varphi}_{(ij)_a} + a_{a-a} \sum_{\substack{(kl)_a \\ k,l \neq i,j}} \left\{\hat{W}^{\theta,\varphi}_{(kl)_a}\right\}_{(ij)_a} \tag{10}$$

Choosing values for all fitting parameters and inserting values for $T_{1e}$ and $T_{2mw}$, the populations of the allowed transitions corresponding to $(\theta,\varphi)_{det}$ can now be obtained using Eq. 10 at the end of a MW pump period $t_{MW}$ at frequency $\nu_{MW}$. The EPR signal $E_{det}(\nu_{det}, t_{MW})$ at $\nu_{det}$ can then be calculated by taking the Hamiltonian diagonalization into account and by solving Eq. 6 with the modified MW rate matrices for each set of angles $(\varphi,\theta)$. Adding all $(p^{\theta,\varphi}_{i_a} - p^{\theta,\varphi}_{j_a})(t_{MW})$ values belonging to $(\theta,\varphi)_{det}$ and normalizing their sum $S_{det}(\nu_{MW}, t_{MW})$ to the sum $S^{ref}_{det}(t_{MW})$ of all $(p^{\theta,\varphi}_{i_a} - p^{\theta,\varphi}_{j_a})(t_{MW})$ belonging to $(\theta,\varphi)_{det}$, obtained by again solving Eq. (10) but this time for a $\nu_{MW}$ value far removed from the frequency range of all allowed and forbidden transitions:

$$E_{det}(\nu_{MW}, t_{MW}) = S_{det}(\nu_{MW}, t_{MW}) / S^{ref}_{det}(t_{MW}) \tag{11}.$$





Plotting $E_{det}(\nu_{MW}, t_{MW})$ as a function of $\nu_{MW}$, and after line smoothing over 5 MHz,
results in a ELDOR spectrum at $\nu_{det}$. (see Fig. 2).
**3.2 High radical concentrations**
To simulate the ELDOR spectra of the 10 mM and 20 mM samples we used the eSD
model (Hovav et al., 2015b). This computational model divides the EPR spectrum into
frequency bins and calculates the electron polarizations $P_b(t_{MW})$ of each bin at
frequency $\nu_b$. It consists of a set of coupled rate equations for these polarizations with
rate constants describing the effects of spin lattice relaxation, eSD polarization
exchange and MW irradiation. To take the SE into account the MW rate constants of
each $P_b(t_{MW})$, are extended by effective SE terms(Hovav et al., 2015b; Kundu et al.,
2018b; Wang et al., 2018):

$$w_{MW}^b = \frac{\omega_1^2 T_{2mw}}{1 + 4\pi^2 (\nu_b - \nu_{MW})^2 T_{2mw}^2} + \sum_{K=H,N,H-N} \frac{(A_K^{SE} \omega_1)^2 T_{2mw}}{1 + 4\pi^2 (\nu_b \pm \nu_K - \nu_{MW})^2 T_{2mw}^2} . \qquad (12)$$

Here $\nu_K$ are the $^1$H and $^{14}$N nuclear frequencies and $A_H^{SE}$, $A_N^{SE}$ and $A_{H-N}^{SE}$ are fitting
parameters used to scale the MW power on the forbidden transition and just affect the
SE peak intensities of the ELDOR peaks and not their positions. The eSD exchange
rate constants between the polarizations in bin $b$ and bin $b'$ are defined by the exchange
rate coefficients

$$r_{b,b'}^{eSD} = \frac{\Lambda^{eSD}}{4\pi^2 (\nu_b - \nu_{b'})^2} , \qquad (13)$$

where the parameter $\Lambda^{eSD}$ determines the time scale of the spectral diffusion process.
After solving the polarization rate equations for an irradiation frequency $\nu_{MW}$ the
polarization $P_{det}(\nu_{MW})$ at the detection frequency $\nu_{det}$ is obtained and divided by its
Boltzman equilibrium value $P_{det}^{eq}$ to obtain the ELDOR signal

$$E(\nu_{MW}, \nu_{det}, t_{MW}) = \frac{P_{det}(\nu_{MW})}{P_{det}^{eq}} \qquad (14).$$





# 4 Results and Discussion

## 4.1 ELDOR spectra of the 0.5 mM TEMPOL

Experimental ELDOR spectra of the 0.5 mM TEMPOL were obtained by recording EPR echo intensities as a function of $\nu_{MW}$ for fixed $\nu_{det}$ and $t_{MW}$ values, using the experimental parameters summarized in the Experimental section. The results $E(\nu_{MW};\nu_{det},t_{MW})$ were analyzed using the procedure described in the Simulation section. From the many ELDOR spectra measured in this way, we show in Fig. 3 (black traces) only three, each one with a different detection frequency $\nu_{det}$ within the EPR spectrum. The dips in the ELDOR spectra, also referred to as EDNMR spectra, appear at the frequencies of the allowed and forbidden transitions, dictated by the $^1$H and $^{14}$N Larmor frequencies $\nu_H$ and $\nu_N$ and their hyperfine interactions $(A_{zz}^H, A_H^{\pm})$ for $^1$H and $(A_{zz}^N, A_N^{\pm})$ for $^{14}$N (Aliabadi et al., 2015; Cox et al., 2013, 2017; Kaminker et al., 2014; Nalepa et al., 2014; Ramirez Cohen et al., 2017; Rapatskiy et al., 2012). At W-band frequencies (~95 GHz) the $^1$H frequencies are around 144 MHz and the $^{14}$N frequencies are in the range : $20-70\,\mathrm{MHz}$, as reported earlier in EDNMR experiments(Florent et al., 2011; Kaminker et al., 2014; Nalepa et al., 2014; Wili and Jeschke, 2018). Thus we expect in addition to the homo-nuclear forbidden transition signals additional signals around -144, 0 and +144 MHz each with a possible spread of -70 - +70 MHz, due to the hetero-nuclear forbidden transitions.

Fig. 3b shows the ELDOR spectrum for $\nu_{det} = 94.55\,\mathrm{GHz}$. This frequency falls in the $g_Z$ region of the EPR spectrum (Fig. 3a), which is characterized by its "single crystal like" features. As a result the $^{14}$N signals are only slightly powder broadened and well resolved.(Florent et al., 2011; Kaminker et al., 2014) At this detection frequency the contributions to the echo signal originate only from the two low frequency allowed transitions (red in the $\Delta\nu_{det} = -250$ MHz stick diagram), split by the $^1$H hyperfine interaction, of the crystallites belonging to the "single crystal". The MW excitation is not selective enough to resolve the protons splitting. In Table S1 in the supplementary the frequency assignments of the lines in the ELDOR spectra are correlated to the



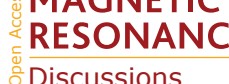

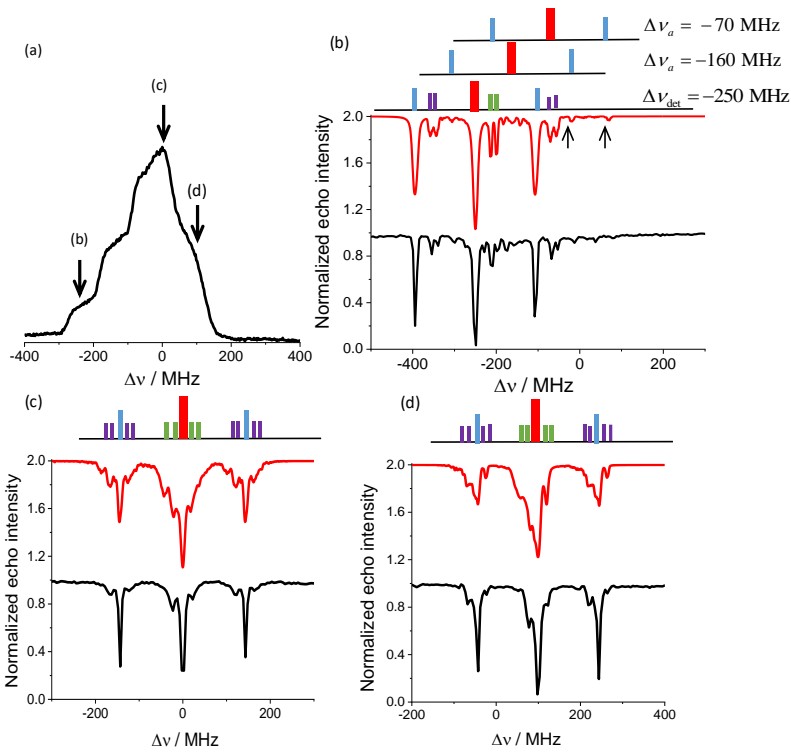

Fig. 3. (d) The EPR spectrum and the positions at which the ELDOR spectra shown in (b)-(d) were recorded. (d)-(d) Experimental (black) and simulated (red) ELDOR spectra along with the associated stick spectrum using the color codes shown in Fig. 2, with detection frequencies $\nu_{det} = 94.55, 94.8, 94.9$ GHz , for (b), (c), and (d) respectively. The frequency axis is plotted relative to the center of the EPR spectrum at 94.8GHz such that $\Delta\nu_{det} = -250$ MHz, 0MHz, 100MHz for (b), (c), and (d) respectively. The (b) spectrum is the most resolved , it shows the $^{14}$N DQ transitions as well as peaks due to the other four allowed transitions and their associated $^1$H forbidden transitions arising from off-resonance and relaxation effects..

$(i-j)_a$ and $(i-j)_f$ transitions in Fig. 2, together with the color coding in the stick
spectrum shown in Fig. 3b. The assignments of the other four allowed transitions are
also tabulated, together with their $^1$H- and $^{14}$N-homonuclear forbidden transitions and
the $^1$H-$^{14}$N-hetreonuclear forbidden transitions. In the ELDOR spectra the two $^1$H-
transitions (in blue) and the four $^{14}$N-transitions (in green) are clearly present. The $^1$H-
$^{14}$N-transitions (in purple) are also detected. The additional spectral features must





originate from the four non-directly detected allowed transitions with their forbidden
transitions. Stick spectra of these allowed transitions and their [1]H-forbidden transitions
are also added in Fig. 3a, and it is interesting to see that part of these lines in these
spectra appear in the experimental ELDOR spectrum (marked by arrows in Fig. 3b).
The appearance of signals corresponding to the no-directly excited  allowed transition
has been reported earlier(Kaminker et al., 2014) and was attributed to the combination
of off-resonance and relaxation effects. In Fig. 3b the experimental ELDOR spectrum
at $\nu_{\text{det}} = 94.8\,\text{GHz}$ $(g_y)$ is plotted and a schematic stick spectrum is added on the top. All
possible allowed transitions contribute to this spectrum and the spectral features are
broadened and even hard to distinguish. The stick spectrum represents only one typical
contribution to the observed powder spectrum. The same is true for the spectrum at
$\nu_{\text{det}} = 94.9\,\text{GHz}$ $(g_x)$ .
To simulate the experimental ELDOR spectra we used the $T_{1e}$ values, which were
measured at several frequency positions within the EPR spectrum:  20.8ms  at
$\nu_{\text{det}} = 94.6\,\text{GHz}$ ,    13.8ms at  $\nu_{\text{det}} = 94.8\,\text{GHz}$ and  15.8ms at  $\nu_{\text{det}} = 94.9\,\text{GHz}$ . Thus the $T_{1e}$
values were found to vary with the position within the EPR spectrum, with the highest
value obtained for the $g_z$ region.  In the simulations we used the average value of
$T_{1e} = 16.7\,\text{ms}$ .
The best fit simulated spectra that resemble the three experimental ELDOR spectra in
Fig. 3 are shown in red. To achieve these spectra we used the following parameters:
$T_{2mw} = 100\,\mu\text{s}$ ,  $t_{MW} = 100\,\text{ms}$  and  the  SE  fitting  parameters  $a_H^{SE} = 10^3$ ,  $a_N^{SE} = 0.5$ ,
$a_{H-N}^{SE} = 10^3$ and $a_{a-a}^{SE} = 0.5 \times 10^{-3}$ . These parameters were determined via manual fitting
of the intensities of the different lines in the spectrum in Fig. 3b. The same parameters
were used for the simulated spectra in Fig. 3c and 3d. The fact that the SE parameter of
the [1]H-forbidden transitions is large, seems to be connected with the many protons
involved in the SE process in the sample. In addition to the above mentioned forbidden
transitions, we added also [14]N double quantum effect in the simulations by introducing
a SE parameter of $a_{DQ}^{SE} = 5$ . The resulting double quantum lines are shown in the
simulated trace in Fig. 3a by small arrows. Comparing the simulated and experimental
spectra we observe all expected forbidden transitions and some lines originating from
the non-observed allowed transitions and their forbidden transitions. The double





quantum lines expected around $\Delta\nu = 200$ MHz are not clearly resolved. The calculated
spectra in Fig. 3c and 3d resemble the experimental spectra, although the relative
intensities of the lines do not agree so well.
A contour plot of the experimental 2D-ELDOR spectrum of the 0.5 mM sample is
shown in Fig. 4a. The positions of the lines corresponding to the allowed transitions
appear at the intense central diagonal of the spectrum. The signals associated with the
{e-$^{14}$N} forbidden transitions are close to the central diagonal and clearly reveal the
anisotropic character of the hyperfine interaction. Namely, the strongest shifts of the
line positions, with respect to the allowed line positions, are about 40 MHz in the $g_z$
region of the EPR spectrum and they become around 20 MHz in the $g_{x,y}$ regime.   The
signals associated with the {e-$^1$H} forbidden transitions are the intense lines parallel to
the diagonal, and are surrounded by the signals coming from the {e-$^1$H-$^{14}$N} forbidden
transitions. Figure 4b shows the simulated 2D-ELDOR contour plot, which reproduces

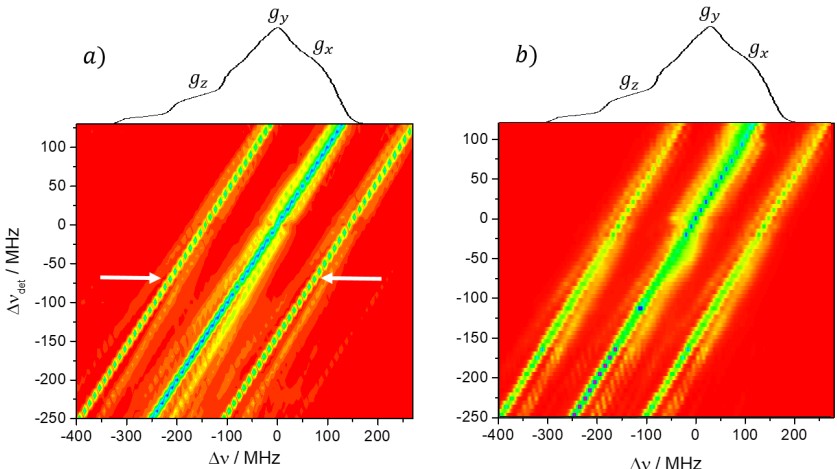

**Figure 4.** 2D contour ELDOR spectra of the 0.5mM sample (a) Experimental and (b) simulated spectra where the y-axis is the off-resonance detection frequency ($\Delta\nu_{det} = \nu_{det} - 94.8$ GHz) and the x-axis is the off-resonance pump frequency ($\Delta\nu_{mw}$). The central diagonal line corresponds to the allowed EPR transitions while the intense parallel lines on both its sides correspond to $^1$H signals as indicated by white arrows in the experimental spectrum. The weaker lines around the center diagonal correspond to forbidden transitions involving $^{14}$N and those about the outer $^1$H lines are due to those involving both $^1$H and $^{14}$N

most of the features observed in the experimental contours. A notable discrepancy is





the weaker $^{14}$N signals on the negative side of the allowed transition, this is presumably
a result of our choice of the co-alignment of the $^1$H and $^{14}$N hyperfine tensors.
**4.2 ELDOR spectra of 10 mM and 20 TEMPOL**
The 2D ELDOR spectrum for a 10 mM TEMPOL solution, presented in Fig. 5, displays
the main features of the $^1$H SE solid effect lines, which run parallel to the diagonal. $^{14}$N
and combination lines are detectable but they are not as nicely resolved as in the 0.5
mM sample. In addition, broad features that correspond to the depolarization of the
electron spins owing to the eSD process are evident. To consider both SE and eSD
effects we simulated the ELDOR spectra using the eSD model, including the influence
of $^{14}$N and $^1$H SE by incorporating the SE features as described in the Simulation section
Eq. 12. We also measured $T_{1e}$ along the EPR spectrum and the results are given in
Fig. 6. $T_{1e}$ displays an anisotropic behavior, namely it depends on the position within
the EPR spectrum with the largest variations observed in the $g_z$ region (similar to our
earlier observation for the 0.5 mM solution). Similar $T_{1e}$ variations was also reported
Weber *et al*(Weber et al., 2017).

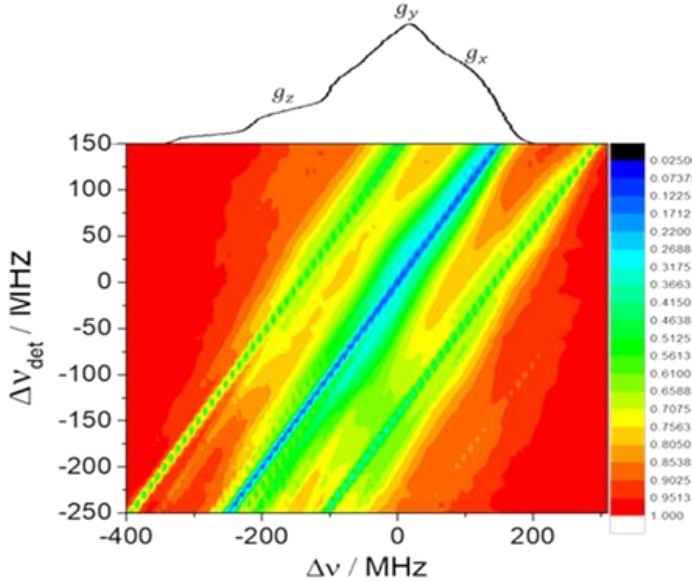

**Figure 5:** Experimental 2D ELDOR spectra of 10 mM TEMPOL solution



To include the experimental $T_{1e}$ values into the simulations, we assigned to each group
of 5 consecutive bins, each one with a width of 2 MHz, the value of $T_{1e}$ measured at
the position in the EPR spectrum that correspond to those bins. Example of
experimental and simulated ELDOR spectra for three positions of the detection
frequency in the EPR spectrum are shown in Figure 7.
Initially the spectra were simulated using the eSD model considering only the $^1$H SE
effect (blue traces in Fig. 7), and the best fit gave an eSD parameter of $\Lambda^{eSD} = 60\ \mu s^{-3}$.
A better fit was obtained when taking into account $^{14}$N SE, including the $^{14}$N-$^1$H
combinations (green traces). This addition broadened the ELDOR lines resulting in a
better match with the experimental result, with the same $\Lambda^{eSD}$ value. Nevertheless,
when $\nu_{det}$ reached the $g_z$ region of the EPR spectrum (Fig. 7a, $\Delta\nu = -100$ MHz), the
fit was not as good as in $g_x$ and $g_y$. This implies that $\Lambda^{eSD}$ might be anisotropic, which
is unexpected. At this point we attribute this "apparent" anisotropy to the over
simplified ad-hoc inclusion of the SE mechanism into the eSD model which does not
fully account for the anisotropy of the $^{14}$N hyperfine interaction. This is probably
insufficient for reproducing the spectral features originating from the $^{14}$N SE
mechanism, which is highly anisotropic and has significant contributions at low
concentrations, throughout the EPR spectrum.

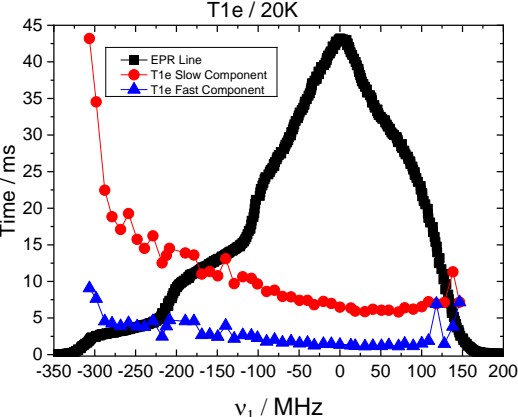

**Figure 6.** The frequency dependence of $T_{1e}$ of 10 mM TEMPOL at 20K, measured every 10 MHz .
Each point corresponds to a measurement fitted with a bi-exponential fit as noted on the figure.

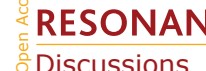

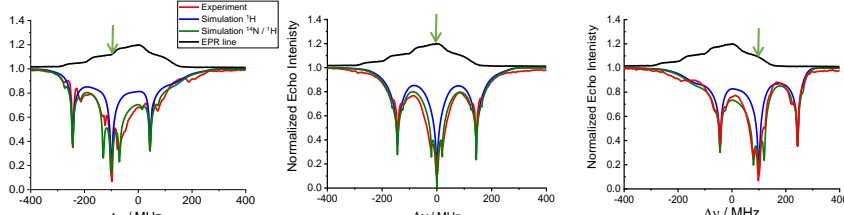

**Figure 7.** Experimental (red) and simulated (blue and green) ELDOR spectra of 10 mM TEMPOL at different positions along the EPR spectrum. All spectra were fitted with $\Lambda^{eSD} = 60 \, \mu s^{-3}$, $T_{1e} = .7ms$, $T_2 = 100 \mu s$. The blue spectra show the result of the simulation including only the $^1$H SE while the green spectra include both $^1$H and $^{14}$N contributions. The detection frequency is marked with a green arrow at the top of each panel. The simulation was performed using 350 frequency bins with a 2 MHz width, spanning the whole EPR spectrum. The pump frequency spanned 1000 MHz with steps of 2 MHz, the forbidden transition fitting parameters were: $A_H^{SE} = 3 \math{g} 0^{-3}, A_N^{SE} = 1.5 \math{g} 0^{-3}, A_{HN}^{SE} = 0.4 \math{g} 0^{-3}$. The NMR frequencies (corresponding the $\nu_\kappa$ in Eq. 12) used in the simulation were $\omega_{H\_NMR} = \pm 144$ MHz, $\nu_{N\_NMR} = \pm 20$ MHz for $^{14}$N, and $\omega_{HN\_NMR} = \omega_H \pm 20$ MHz for the $^1$H and $^{14}$N combinations.

To examine the degree of the influence of the $^{14}$N SE on the electron depolarization at
higher radical concentrations, where the ELDOR spectrum is shaped primarily by the
eSD process, we tested also the 20 mM sample and used the eSD model to simulate the
ELDOR lineshape recorded with $\nu_{det}$ set to the center of the EPR spectrum, as shown
in Figure 8. Because of the high electron spin concentration, the eSD causes large
depolarization of the EPR spectrum, which translates in extensive broadening of the
ELDOR spectrum.
Figure 8 shows in red the experimental ELDOR spectrum, where although the lineshape
of this spectrum is determined by the eSD process, we can still see small signals coming
from the $^{14}$N SE. Simulation including both the $^1$H and $^{14}$N SE with $\Lambda^{eSD} = 400 \mu s^{-3}$
gave a good agreement with the experimental spectrum. In contrast, setting $\Lambda^{eSD} =$
$400 \mu s^{-3}$ and taking into account only the contributions of the $^1$H SE, did not result in a
good fit. This shows that even at relative high radical concentrations, the effect of the





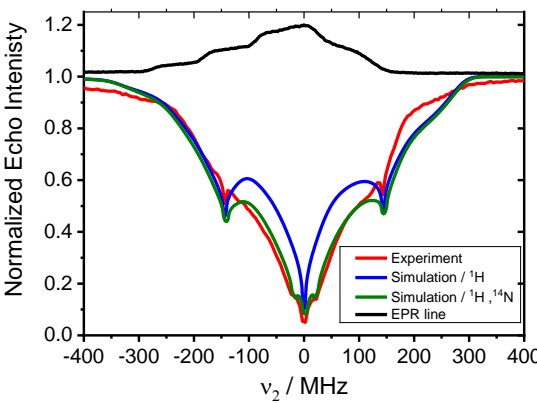

**Figure 8.** Experimental (red) and simulated (blue and green) ELDOR spectra of 20 mM TEMPOL recorded at the maximum of the EPR spectrum. The fit was achieved with $\Lambda^{eSD} = 400$ μs,$^{-3}$ $T_{1e} = 5.7 ms$, $T_2 = 100 \mu s$. The blue spectra show the result of the simulation including the only the $^1$H SE while the green spectra include both $^1$H and $^{14}$N contributions. The forbidden transition fitting parameters were: $A_H^{SE} = 3$g$0^{-3}$, $A_N^{SE} = 5$g$0^{-3}$, $A_{HN}^{SE} = 0.4$g$0^{-3}$ and the nuclear frequencies were the same as in Fig. 7.

depolarization due to the $^{14}$N SE can still be significant and if not included can introduce
inaccuracies in the eSD parameters and thus also in the DNP spectra, derived from the
EPR lineshapes that are constructed using these parameters. Earlier measurements[7]
showed that 20 mM TEMPOL concentration, ELDOR spectra measured at the $g_y$ and
$g_z$ position gave the same quality fit with the same $\Lambda^{eSD}$, implying that at this
concentration the relative contribution of the $^{14}$N SE mechanism is small and can be
accounted for by the simple model presented in this work.
# 5 Conclusions
In this work we focused on the contributions of the $^{14}$N SE to the depolarization gradient
within the EPR spectrum of TEMPOL, during long MW irradiation, as commonly used
in DNP measurements, as determined by ELDOR measurements. For low concentration
(0.5 mM) TEMPOL samples, where the SE dominates and eSD is negligible, we have
successfully reproduced all the SE related depolarization signals, including those
involving combinations of $^1$H-$^{14}$N associated forbidden EPR transitions. Subsequently,



we used the eSD model (Hovav et al., 2015c)(Hovav et al., 2015c)(Hovav et al.,
2015c)(Hovav et al., 2015c) to simulate ELDOR spectra of 10 and 20 mM TEMPOL
samples with ad-hoc addition of electron depolarization due to the $^{14}$N SE based on the
frequencies determined from the 0.5 mM sample. We observed that simulations
including the $^{14}$N SE improved the fit with experimental ELDOR spectra for the 10 mM
sample. However we notice that at the $g_z$ region of the EPR spectrum the fit is not as
good indicating that the model is still not good enough to take into account the large
$^{14}$N SE contributions in this region. For the 20 mM concentration the effect is more
drastic affecting significantly the best fitted value of $\Lambda^{eSD}$ . We conclude that including
$^{14}$N SE in the eSD model is essential for obtaining reliable fitting at high radical
concentrations.
Acknowledgments
This work is supported by a grant of the Binational Science Foundation (Grant #2014149)
and was made possible in part by the historic generosity of the Harold Perlman Family (D.
G.). D. G. holds the Erich Klieger Professorial Chair in Chemical Physics.

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
