# Peer review of "Study of electron spectral diffusion process under DNP conditions by"

_Magnetic Resonance, 2020_

## Referee Comment (RC1) · Anonymous Referee #1 · 4 Mar 2020

This manuscript reports on the influence of the $^{14}$N solid effect on electron spectral diffusion profiles measured at nitroxide radical concentrations relevant for dynamic nuclear polarization. The authors approach the problem by combining considerations on spin dynamics with an empirical parametrized fitting model. This approach leads to an improvement compared to simulations disregarding the $^{14}$N solid effect and, indeed, at 20 mM concentration to reasonable agreement with experimental results. At an intermediate concentration of 10 mM, the model turns out to be simplistic. This is useful work, which improves understanding of electron spectral diffusion at high nitroxide radical concentrations. I recommend publication in Magnetic Resonance after minor revision that takes into account the following suggestions:

[Figure]
1. It is very awkward to report and use a nitroxide g tensor g = [2.0065, 2.0037, 1.9997], which is certainly wrong. The values reported earlier by Florent et al. were (almost) in line with expectations from numerous other reports and from quantum-chemical computations. The sentence quoting a "systematic error of 4 mT in the determination of the external magnetic field" is ambiguous, as it does not tell whether the error applies to Florent et al. or to the present work (it is the present work). You should be able to check this by using the proton Larmor frequency from ELDOR-detected NMR for calibration. It may not be necessary to repeat all computations (the anisotropy is correct, the absolute error too small to influence these simulations), but the issue should be reported in a clear way.

2. Please explain why $T_M$ can serve as a lower limit for $T_{2mw}$ (page 11, line 13, and wouldn't $T_{2\rho}$ be a better choice?). This is not obvious to me.

3. You claim (page 19, line 2) that the weaker $^{14}$N signals on the negative side of the allowed transition are presumably due to co-alignment of $^{14}$N and $^1$H hyperfine tensors. Did you test this?

4. You appear to cite every paper on W-band ELDOR-detected NMR on nitroxides (page 4, line 7/8), except for the very first one (DOI: 10.1016/S0009-2614(98)00765-9)

Typos:

Page 4, line 29: space missing between "0.6" and "mm"

Page 9, caption Figure 2: there is a surplus red "[** small m]"

Page 11, line 24: "signals reveal very weak signals" is awkward (signals are very weak)

Page 12, line 7: "three-spin calculation" (delete surplus "s")

Page 12, line 15: "we added reintroduced" should only read "we added"

Page 12, line 20: "on for the different forbidden transitions" shouldn't this read "one for each of the forbidden transitions"?

Page 14, line 2: "see Fig. 3" (not 2)

Page 15, line 15: notorious Microsoft Word box for a symbol in front of "20-70 MHz"

Page 16, caption Fig. 3: spaces missing between "0" and "MHz" as well as between "100" and "MHz"

Page 22, line 3: Reference with superscript 7 should be in Name, Year format

---

## Short Comment (SC1) · 9 Mar 2020

A very short comment regarding the sentence on page 4:

"Finally, so far contributions from different nuclei in the EDNMR spectra were taken into account by superimposing their individual spectra(Wang et al., 2018), ignoring the contributions of combination frequencies(Tan et al., 2019). Here we also account for 14N-1H combinationlines in the ELDOR spectrum."

Combination lines in ELDOR-detected NMR were already included in the work of (Cox et al., 2017) (https://doi.org/10.1016/j.jmr.2017.04.006), although mainly for deuterium

and nitrogen.

---

## Referee Comment (RC2) · Anonymous Referee #2 · 11 Mar 2020

This paper discusses experimentally and theoretically the ELDOR spectrum of TEM-POL solutions in presence of protons and 14N nuclei. Three different TEMPOL concentrations are considered : • At very low concentrations the electron spin of TEMPOL are weakly interacting and the ELDOR spectrum can be computed considering a collection of randomly oriented 3 spin systems (one electron and two nuclei) • At very high concentration the ELDOR spectrum is dominated by the electron-electron interaction. A phenomenological model that mimics the spectral diffusion between different electrons reproduce well the experimental data • At intermediate concentration the interplay between spectral diffusion and depolarization induced by 14N solid effect is not correctly described by the model. The paper contains a large amount of work,
is well written and should be published. I have few comments : 1) I was not able to clearly find the temperature at which the experiments are performed (20 Kelvin ?). 2) In the introduction the papers Kundu 2018a and Kundu 2018b are cited because they performed quantum mechanical based calculations of the EPR spectra. They found a connection between eSD models and thermal mixing regime. The paper Caracciolo et al (PCCP , 2016, vol. 18, no 36, p. 25655-25662.) should also be cited in this context. 3) Before equation 7c the thermal equilibrium populations should be specified (they are not in the rotating frame)

---

## Short Comment (SC2) · 20 Mar 2020

see attached

Please also note the supplement to this comment:
https://www.magn-reson-discuss.net/mr-2020-3/mr-2020-3-SC2-supplement.pdf

---

## Short Comment (SC3) · 20 Mar 2020

Thank you Nino for pointing this out – apologies for the omission. We revised this part and now we write in p. 5 In some earlier works the contributions from different nuclei in the EDNMR spectra were taken into account by superimposing their individual spectra ignoring the contributions of combination frequencies (Tan et al., 2019, Wang et al., 2018). In others, the combinations were also taken into account and reproduced in the simulated spectra (Cox et al., 2017). The appearance of these lines depends on the experimental conditions (Cox et al., 2017). As under DNP conditions the duration of the microwave irradiation is long we also took into account for 14N-1H combination

lines in the ELDOR spectral simulations.

---

## Short Comment (SC4) · 20 Mar 2020

Thank you reviewer 2.   Below we provide a response to your comments.  Our response is in red and the uploaded revised manuscript highlights  the changes made.

**Anonymous Referee #2**

This paper discusses experimentally and theoretically the ELDOR spectrum of TEMPOL solutions in presence of protons and 14N nuclei. Three different TEMPOL concentrations are considered : ??? At very low concentrations the electron spin of TEMPOL are weakly interacting and the ELDOR spectrum can be computed considering a collection of randomly oriented 3 spin systems (one electron and two nuclei) ???. At very high concentration the ELDOR spectrum is dominated by the electron-electron interaction. A phenomenological model that mimics the spectral diffusion between different electrons reproduce well the experimental data ??? At intermediate concentration the interplay between spectral diffusion and depolarization induced by 14N solid effect is not correctly described by the model. The paper contains a large amount of work,

is well written and should be published. I have few comments :

1) I was not able to clearly find the temperature at which the experiments are performed (20 Kelvin ?).

   Yes, this is mentioned at the experimental section under Spectroscopic measurements. We added in each Fig. caption the temperature to avoid confusion.

2) In the introduction the papers Kundu 2018a and Kundu 2018b are cited because they performed quantum mechanical based calculations of the EPR spectra. They found a connection between eSD models and thermal mixing regime. The paper Caracciolo et al (PCCP , 2016, vol. 18, no 36, p. 25655-25662.) should also be cited in this context.

This was added in p. 2.

3) Before equation 7c the thermal equilibrium populations should be specified (they are not in the rotating frame)

The sentence was modified in p. 11 to:

" …being the ratio between thermal equilibrium populations defined in the laboratory frame, and "

---

## Short Comment (SC5) · 23 Mar 2020

see the attached revised manuscript with the highlighted changes.

Please also note the supplement to this comment:
https://www.magn-reson-discuss.net/mr-2020-3/mr-2020-3-SC5-supplement.pdf

---

## Editor Comment (EC1) · Konstantin Ivanov (Editor) · 23 Mar 2020

Dear Prof. Goldfarb, dear Prof. Vega,

The period for open discussion of your paper "Study of electron spectral diffusion process under DNP conditions by ELDOR spectroscopy focusing on the 14N Solid Effect" has expried. You have recieved comments from two referees, which are positive, suggesting minor revision of your paper.

As I can see, you have already provided responses to both reviewers. I request you to submit the revised paper as well. I will have a look at the changes you have made

in the manuscript, if necessary the paper will go through an additional (swift) round of refereeing (carried out by one of the original reviewers).

Sincerely yours,

Konstantin Ivanov.
* * *

---

## Author Comment (AC2) · 25 Mar 2020

The comment was uploaded in the form of a supplement:
https://www.magn-reson-discuss.net/mr-2020-3/mr-2020-3-AC2-supplement.pdf

---

## Author Response (AR1)

Thank you to the reviewers.  Below we provide a response to your comments.  Our response is in red and the uploaded revised manuscript highlights  the changes made.

**Anonymous Referee #1**

This manuscript reports on the influence of the 14N solid effect on electron spectral diffusion profiles measured at nitroxide radical concentrations relevant for dynamic nuclear polarization. The authors approach the problem by combining considerations on spin dynamics with an empirical parametrized fitting model. This approach leads to an improvement compared to simulations disregarding the 14N solid effect and, indeed, at 20 mM concentration to reasonable agreement with experimental results. At an intermediate concentration of 10 mM, the model turns out to be simplistic. This is useful work, which improves understanding of electron spectral diffusion at high nitroxide radical concentrations. I recommend publication in Magnetic Resonance after minor revision that takes into account the following suggestions:

1. It is very awkward to report and use a nitroxide g tensor g = [2.0065, 2.0037, 1.9997], which is certainly wrong. The values reported earlier by Florent et al. were (almost) in line with expectations from numerous other reports and from quantum-chemical computations. The sentence quoting a "systematic error of 4 mT in the determination of the external magnetic field" is ambiguous, as it does not tell whether the error applies to Florent et al. or to the present work (it is the present work). You should be able to check this by using the proton Larmor frequency from ELDOR-detected NMR for calibration. It may not be necessary to repeat all computations (the anisotropy is correct, the absolute error too small to influence these simulations), but the issue should be reported in a clear way.

We agree with the reviewer, indeed the reported g values are off and this is because of an error in the determination of the magnetic field. Unfortunately the EDNMR spectra were recorded with a too low resolution (because of the wide spectral width covered) to track this.  In retrospect, the field should have been calibrated with a high resolution EDNMR in the $^1$H region when the measurements were done.  The shift for the proton frequency is 0.17 MHz, which is very small compared to linewith observed. For the $^{14}$N it is smaller and therefore small error in the  nuclei Larmor frequencies are negligible. Similarly, as the anisotropy of **g**  is correct this small shift does not affect the selected orientation and the other terms that depend on **g**  because the energies and their differences depend on the product $g_{eff} \times B$, where the error in B has been compensated in **g**. We now explain this in the manuscript in p. 7 as follows:

The *g*-values obtained from the EPR simulations and further used in the EDNMR simulations  differ from those reported by Florent *et al* (Florent et al., 2011) (*g* = [2.00988, 2.00614, 2.00194]) as they compensate for an error of 4 mT in determination of $B_0$. These g-values where used to determine the selected orientations and to calculate $g_{ef}$ in Eq. 2. Because the energies and their differences depend on the product $g_{eff}B_0$, where the error in $B_0$ has been compensated in **g**, they are not affected by the error in the field. The shift of 4 mT in $B_0$ for the proton frequency 0.17 MHz, which is very small compared to EDNMR linewidth. For $^{14}$N it is smaller and therefore small error in the  nuclei Larmor frequencies are negligible.

2. Please explain why TM can serve as a lower limit for T2mw (page 11, line 13, and

wouldn't T2$\rho$ be a better choice?). This is not obvious to me.

You are correct, but we did not measure T$_2\rho$ and we deleted this part of the sentence to avoid confusion.

3. You claim (page 19, line 2) that the weaker 14N signals on the negative side of the allowed transition are presumably due to co-alignment of 14N and 1H hyperfine tensors. Did you test this?

This was a misplaced sentence and it is incorrect. We revised this part and wrote Figure 4b shows the simulated 2D-ELDOR contour plot, which reproduces most of the features observed in the experimental contours. Some discrepancies can be observed in the intensities of the forbidden transition lines which can be attributed to the simplifications of the model.

4. You appear to cite every paper on W-band ELDOR-detected NMR on nitroxides (page 4, line 7/8), except for the very first one (DOI: 10.1016/S0009-2614(98)00765-9)

We apologize for this unintentional omission and added the reference in p. 4.

Typos:
Page 4, line 29: space missing between "0.6" and "mm"
Page 9, caption Figure 2: there is a surplus red "[** small m]"
Page 11, line 24: "signals reveal very weak signals" is awkward (signals are very weak)
Page 12, line 7: "three-spin calculation" (delete surplus "s")
Page 12, line 15: "we added reintroduced" should only read "we added"
Page 12, line 20: "on for the different forbidden transitions" shouldn't this read "one for each of the forbidden transitions"?

Thanks  - we fixed all these.

**Anonymous Referee #2**

This paper discusses experimentally and theoretically the ELDOR spectrum of TEMPOL solutions in presence of protons and 14N nuclei. Three different TEMPOL concentrations are considered : ??? At very low concentrations the electron spin of TEMPOL are weakly interacting and the ELDOR spectrum can be computed considering a collection of randomly oriented 3 spin systems (one electron and two nuclei) ???. At very high concentration the ELDOR spectrum is dominated by the electron-electron interaction. A phenomenological model that mimics the spectral diffusion between different electrons reproduce well the experimental data ??? At intermediate concentration the interplay between spectral diffusion and depolarization induced by 14N solid effect is not correctly described by the model. The paper contains a large amount of work,

is well written and should be published. I have few comments :

1) I was not able to clearly find the temperature at which the experiments are performed (20 Kelvin ?).

Yes, this is mentioned at the experimental section under Spectroscopic measurements. We added in each Fig. caption the temperature to avoid confusion.

2) In the introduction the papers Kundu 2018a and Kundu 2018b are cited because they performed quantum mechanical based calculations of the EPR spectra. They found a connection between eSD models and thermal mixing regime. The paper Caracciolo et al (PCCP , 2016, vol. 18, no 36, p. 25655-25662.) should also be cited in this context.

This was added in p. 2.

3) Before equation 7c the thermal equilibrium populations should be specified (they are not in the rotating frame)

The sentence was modified in p. 11 to:

" …being the ratio between thermal equilibrium populations defined in the laboratory frame, and "